# Insulin-like Androgenic Gland Hormone Induced Sex Reversal and Molecular Pathways in *Macrobrachium nipponense*: Insights into Reproduction, Growth, and Sex Differentiation

**DOI:** 10.3390/ijms241814306

**Published:** 2023-09-19

**Authors:** Pengfei Cai, Wenyi Zhang, Sufei Jiang, Yiwei Xiong, Huwei Yuan, Zijian Gao, Xuanbing Gao, Cheng Ma, Yongkang Zhou, Yongsheng Gong, Hui Qiao, Shubo Jin, Hongtuo Fu

**Affiliations:** 1Wuxi Fisheries College, Nanjing Agricultural University, Wuxi 214081, China; ckgg5436@126.com (P.C.); yuan08102021@126.com (H.Y.); gaozijiangenomics@163.com (Z.G.); gaoxuanbin@163.com (X.G.); 18225489231@163.com (C.M.); 18555601471@163.com (Y.Z.); 2Key Laboratory of Freshwater Fisheries and Germplasm Resources Utilization, Ministry of Agriculture and Rural Affairs, Freshwater Fisheries Research Center, Chinese Academy of Fishery Sciences, Wuxi 214081, China; zhangwy@ffrc.cn (W.Z.); jiangsf@ffrc.cn (S.J.); xiongyw@ffrc.cn (Y.X.); gongys@ffrc.cn (Y.G.); qiaoh@ffrc.cn (H.Q.)

**Keywords:** *Macrobrachium nipponense*, sex reversal, *IAG*, testis, ovary, reproduction, growth regulation, transcriptome, photoperiodic pathways

## Abstract

This study investigated the potential to use double-stranded RNA insulin-like androgenic gland hormone (*dsIAG*) to induce sex reversal in *Macrobrachium nipponense* and identified the molecular mechanisms underlying crustacean reproduction and sex differentiation. The study aimed to determine whether *dsIAG* could induce sex reversal in PL30-male *M. nipponense* during a critical period. The sex-related genes were selected by performing the gonadal transcriptome analysis of normal male (dsM), normal female (dsFM), neo-female sex-reversed individuals (dsRM), and unreversed males (dsNRM). After six injections, the experiment finally resulted in a 20% production of dsRM. Histologically, dsRM ovaries developed slower than dsFM, but dsNRM spermathecae developed normally. A total of 1718, 1069, and 255 differentially expressed genes were identified through transcriptome sequencing of the gonads in three comparison groups, revealing crucial genes related to reproduction and sex differentiation, such as *GnRHR*, *VGR*, *SG*, and *LWS*. Principal Component Analysis (PCA) also distinguished dsM and dsRM very well. In addition, this study predicted that the eyestalks and the “phototransduction-fly” photoperiodic pathways of *M. nipponense* could play an important role in sex reversal. The enrichment of related pathways and growth traits in dsNRM were combined to establish that *IAG* played a significant role in reproduction, growth regulation, and metabolism. Finally, complete sex reversal may depend on specific stimuli at critical periods. Overall, this study provides valuable findings for the *IAG* regulation of sex differentiation, reproduction, and growth of *M. nipponense* in establishing a monoculture.

## 1. Introduction

The androgenic gland (AG) is a distinct endocrine gland found in male crustaceans, situated near the base of the fifth pereiopod [1]. It plays a crucial role in regulating male sexual differentiation and maintaining secondary sexual characteristics. Insulin-like androgens (*IAG*s) are thought to be the active substances responsible for stimulating the AG to encode and secrete many androgens [2]. Recent studies have conducted experiments on various decapods such as lobsters [3], crayfish [4,5,6], shrimps [7,8,9,10], and crabs [11,12,13,14,15]. One noteworthy study by Ventur et al. successfully achieved functional sex reversal using *IAG* dsRNA (*dsIAG*) in *Macrobrachium rosenbergii* [16]. The spermatophore attached by the male onto the ventral cephalothorax of the *IAG*-silenced males (neo-females) was maintained. They then laid eggs, and all-male offspring were hatched in their brood chambers. The offspring produced from new females and normal males were reported to be male-only, and these second-generation males exhibited faster growth compared to normal males [17]. However, it was difficult to achieve sex reversal successfully by silencing *IAG*. A 2009 study observed that *dsIAG* injections only resulted in the degeneration of male traits [18]. Therefore, factors such as injection cycle, concentration, and environment can affect the success of sex reversal even for the same gene in the same species.

The oriental river prawn (*Macrobrachium nipponense*) [19,20] is a sexually dimorphic crustacean widely distributed in China, Japan, and Korea, with a value of over 20 billion dollars annually [21]. However, during its reproductive period, adult female *M. nipponense* experience rapid and cyclic precocious gonadal maturation that leads to a significant decrease in market size. To address this issue, we used *dsIAG* to artificially control sex and develop monosex cultivation techniques. Previous studies have indicated that crustaceans possess sex plasticity, but it was a challenge to induce sex reversal when the developmentally sensitive period passed [22,23]. Sex formation in crustaceans involves sex determination and sex differentiation. The gonads of *M. nipponense* begin to develop at PL10 (PL, post-larvae developmental stage), sexual differentiation is completed at PL25 with the emergence of physiological males and females, and secondary sexual characteristics gradually appear at PL30 (Appendix A) [24]. A sequence of genes for sex identification has been recognized in *M. rosenbergii* [16]. However, this sequence is not applicable to *M. nipponense*. Therefore, determining whether sex reversal is occurring can be performed by distinguishing secondary sex characteristics. PL30 male prawns are injected five times and then observed to determine whether the experimental group produces female prawns.

In *M. nipponense*, a male reproductive endocrine axis has been identified between the X-organ-sinus gland (XO-SG), AG, and testis. The XO-SG located in the eyestalks is reported to store and release neurohormones such as hyperglycemic hormone (*CHH*), gonadal inhibitory hormone (*GIH*), and molting inhibitory hormone (*MIH*) [25]. AG becomes larger after surgery to remove both eyestalks, and this increase in size is accompanied by the growth of cells that produce *IAG* [11,26,27]. However, *IAG* is a key factor in sex differentiation. This suggests that the existence of a sex-determining pathway upstream of *IAG* probably originates from the eyestalks and influences the testis and AG by regulating *IAG*. In addition, it has been reported that *IAG* inhibits sperm production in males and prevents testis development, accompanied by AG hypertrophy and reduced growth traits [18]. Male prawns that do not reverse sex may convert their energy from reproduction into growth through *IAG*.

In this study, we successfully reversed the sex of PL30 male *M. nipponense* using *dsIAG* and established gonadal transcriptome libraries for neo-females (dsRM) and unsex-reversed males (dsNRM). The objectives of this study were to (1) determine whether *dsIAG* could induce sex reversal in male *M. nipponense*; (2) compare histological and transcriptional differences in the gonads of dsRMs to reveal the mechanism by which *dsIAG* reverses between males and females, and find the upstream metabolic pathways regulating *IAG*; and (3) compare histological and transcriptional differences in the testes of unsex-reversed males, and analyze the response of reproduction-related genes and pathways to *IAG* regulation. The results of this study provide significant insights and methodologies for achieving sex reversal in *M. nipponense*, laying the groundwork for identifying the mechanisms of reproduction and sex differentiation in crustaceans.

## 2. Results

### 2.1. Effects of dsIAG on Male PL30 Juvenile Prawn

#### 2.1.1. Interference Efficiency and Sex Ratio

Figure 1A shows the expression of *MnIAG* in male prawns on days 1, 7, 14, 20, 25, and 30 post-injection. The expression of *MnIAG* in the experimental group was always significantly lower than that in the control group (*p* < 0.05), with percentages of 21.36, 7.68, and 12.32% relative to the control group on days 20, 25, and 30, respectively. The percentages of the male on different days after the injection of PL30 prawns is shown in Figure 1B. RNAi showed a significant decrease in male percentage as the number of injections increased (*p* < 0.05). Moreover, the experimental group consistently had a lower percentage than the control group. Approximately 20% of neo-female prawns were present in the experimental group after six injections.

#### 2.1.2. Histological Observations of the Gonad

Figure 2 shows the gonadal sections of males (dsM), females (dsFM), unsex-reversed males (dsNRM), and neo-females (dsRM). Male and female prawns in the control group developed normally (refer to Figure 2A,C). The male prawns had abundant primary and secondary spermatocytes, as well as sperm, stored in their spermathecae. Meanwhile, the ovaries of the female prawns progressed to stage II, displaying oogenic and follicular cells, and nuclei were observed. Primary and secondary spermatocytes were observed in dsNRM, with no discernible differences compared to dsM. Compared with dsFM, dsRM showed many oogenic and follicular cells, with more abundant yolk granules and more mature ovarian development.

#### 2.1.3. Growth Traits

Table 1 shows the effect of *dsIAG* on the growth traits of juvenile prawns at PL30. This study measured the body weight of the two groups of prawns. The control group had initial weights of 129.85 ± 1.90 mg for males and 141.41 ± 1.98 mg for females. After 30 days, dsM weighed 262.57 ± 9.49 mg and dsFM weighed 353.37 ± 19.52 mg. The RNAi group comprised male prawns with an initial weight of 112.32 ± 2.64 mg. The weight of dsRM after six injections was 265.40 ± 17.28 mg, and the weight of dsNRM was 305.93 ± 6.82 mg. Unsex-reversed males had a significantly higher weight, weight growth rate (WGR), and specific growth rate (SGR), when compared with those of dsM (*p* < 0.05). However, neo-females had a slightly lower weight than normal females, with no significant differences observed in WGR and SGR.

### 2.2. The Comparative Transcriptomic Analysis

#### 2.2.1. Overview of Transcriptome Sequencing

Gonad samples were collected from both control and RNAi groups on day 30 after *dsIAG* injection at PL30. The dsM, dsFM, dsRM, and dsNRM were selected for transcriptome sequencing. After filtering out low-quality reads using FastQC, the four groups yielded the following data: dsM with 6,807,376,125 clean reads, dsFM with 6,764,597,475, dsRM with 6,508,733,250, and dsNRM with 6,303,738,825 (Table 2). All samples passed the FastQC quality check with Q20 values over 95% and Q30 values over 90%, indicating a high sequencing quality. The sequencing reads were deposited in the Short Read Archive (SRA) at the National Center for Biotechnology Information (NCBI) and can be accessed through accession number PRJNA976798.

#### 2.2.2. Identification and Functional Analysis of DEGs

Principal components analysis (PCA) was used to calculate the correlation coefficient between different samples and differentiate them from one another. The resulting PCA clustering can be seen in Figure 3. These analyses provide a powerful tool for understanding the relationships between different samples and their underlying gene expression profiles. During the analysis, the original *p*-values resulting from hypothesis testing were corrected using the widely accepted and effective Benjamini–Hochberg method. This approach allowed for the reduction in the False Discovery Rate (FDR) when screening differentially expressed genes by calculating adjusted *p*-values. The screening criteria used to identify significant DEGs were FC ≥ 2 and FDR ≤ 0.01.

This study found 1718 DEGs in the dsM vs. dsFM comparison, of which 1426 were upregulated and 292 were downregulated. The dsM vs. dsRM comparison revealed 1069 DEGs, with 736 upregulated and 333 downregulated. In the dsFM vs. dsNRM comparison, 255 DEGs were identified, consisting of 135 upregulated genes and 120 downregulated genes. After being annotated by Swiss-Prot, GO, KEGG, COG, KOG, Pfam, and NR, the gene overlap between dsM vs. dsFM and dsM vs. dsRM was 450 genes, whereas dsFM vs. dsNRM shared 70 genes in Figure 4.

#### 2.2.3. GO and COG Enrichment Analysis of DEGs

The GO and COG databases were used to cluster gene products based on their functionalities. The abscissa is the second-level term under the three categories of GO. Unigenes matched with known proteins in the GO database were classified into cellular components (40,539 unigenes), molecular functions (57,149 unigenes), and biological processes (79,990 unigenes) and are presented in Figure 5. Within cellular components, the majority of genes were enriched in the nucleus (4899 unigenes), extracellular region (3229 unigenes), and integral component of the membrane (2924 unigenes). For molecular functions, RNA-directed DNA polymerase activity (6086 unigenes), nucleic acid binding (6088 unigenes), and endonuclease activity (5941 unigenes) were predominant. Biological processes were mainly represented by DNA integration (6060 unigenes), transposition, DNA-mediated (1906 unigenes), and DNA-templated (1098 unigenes). A total of 11,994 unigenes were assigned to the matched proteins in the COG database and included 23 functional categories (Figure 6). The categories with the highest number of assigned unigenes were replication, recombination, and repair (6083 unigenes), followed by general function prediction only (1295 unigenes).

#### 2.2.4. KEGG Analysis and Important Differentially Expressed Pathways

The unigenes were analyzed using KEGG to identify relevant biological pathways, which were divided into five categories: cellular processes, environmental information processing, genetic information processing, metabolism, and organismal systems. Figure 7A shows that 111 DEGs mapped to 77 pathways in the comparison of dsM vs. dsRM. Figure 7B shows that 31 DEGs were enriched across 44 pathways in the comparison of dsFM vs. dsNRM.

#### 2.2.5. Key Pathways and Genes for Reproduction in *M. nipponense*

Table 3 presents significant qvalue differences between dsM vs. dsFM and dsM vs. dsRM in four pathways, including phototransduction-fly, Hippo signaling pathway-fly, phagosome, and ECM-receptor interaction. Table 4 presents significant differences in q-values between dsM vs. dsFM and dsFM vs. dsNRM in four pathways, including lysosome, steroid biosynthesis, thiamine metabolism, and longevity-regulating pathway-multiple species.

A total of 18 reproduction-related genes were screened in the three comparison groups, including 14 sex-related genes and 4 growth-related genes. As shown in Table 5, the genes upregulated in dsM vs. dsFM were vitellogenin, vitellogenin receptor, VASA-like, cyclin B, Fem1b, ferritin, gonadotropin-releasing hormone receptor (*GnRHR*), cystatin, fatty acid synthase, and acetyl-CoA carboxylase, while the downregulated genes were doublesex and the mab-3 related transcription factor, and heat shock protein. Some additional genes that occurred in dsM vs. dsRM were sperm gelatinase, Kazal-type protease inhibitor, male reproductive-related protein, long wavelength-sensitive opsin, and delta-9 desaturase. Glutathione S-transferase was specifically upregulated when dsM vs. dsFM was compared with dsM vs. dsRM.

#### 2.2.6. Validation of DEGs by qRT-PCR

For the evaluation of sequencing and data analysis, nine differentially expressed genes (DEGs) were chosen at random from the transcriptome results for qRT-PCR analysis. Figure 8 shows the results of the qRT-PCR analysis, with positive and negative numbers indicating upward and downward trends, respectively. The expression patterns of these DEGs generally matched those obtained using RNA-Seq analyses, but the relative expression levels were not entirely consistent. Nevertheless, this result confirmed the reliability of the transcriptome sequencing data.

## 3. Discussion

This study aimed to determine the potential of *dsIAG* for sex reversal in *M. nipponense* and to identify the molecular mechanisms of crustacean reproduction and sex differentiation through the transcriptomic analysis of neo-females and unsex-reversed males. We investigated the efficacy of *dsIAG* injection on PL30 male *M. nipponense* during the critical period of sex differentiation by observing the sex ratio and gonadal sections after 30 days of treatment, resulting in a 20% production of sex-reversed individuals (neo-females). The RNAi efficiency showed that the experimental group sustained the suppression of *MnIAG* expression over six injections. Histological sections revealed no significant differences between neo-females and dsNRM, when compared to normal dsM and dsFM prawns, with mature sperm in dsNRM and more yolk granules accumulated in dsRM, implying that there was an optimal period for sex reversal. In a previous report, we fed *M. nipponense* with 17β-estradiol (E_2_) and found that after a certain period [22], even high levels of hormones were unable to alter their sex. *IAG* is responsible for encoding androgens, so we speculate that the sex reversal period in *M. nipponense* may be influenced by androgen levels in males.

In this study, neo-females are smaller in body size compared to normal female prawns. The gonads of dsM, dsFM, dsRM, and dsNRM were analyzed using transcriptome sequencing to characterize transcriptional regulation changes during sex reversal. A total of 62,875 transcripts were obtained, of which the numbers of DEGs for dsM vs. dsFM, dsM vs. dsRM, and dsFM vs. dsNRM were 1718, 1069, and 255, respectively. According to Figure 3, dsM, dsRM, and dsNRM can be clearly distinguished, which is another evidence of the success of sex reversal. GO analysis indicated that reproduction and sex differentiation-related genes were predicted to be mainly found in RNA-directed DNA polymerase activity, nucleic acid binding, and DNA integration. Using the COG classification, these genes primarily occurred in “replication, recombination, and repair” and “general function prediction only” categories. Verification of the RNA-Seq results was confirmed through qPCR analysis of nine randomly selected DEGs, showing congruent expression patterns.

Light has a significant impact on the biological clock and activity of animals [28]. It provides the organism with energy and also serves as an environmental signal to regulate metabolic and reproductive functions [29]. The photoperiodic pathway in vertebrates controls the release of thyrotropic hormones, which then regulate the release of gonadotropin-releasing hormone [30]. In invertebrates, photoreceptors in insects also play key roles in mating and other specific behaviors [30]. Khalaila et al. (2003) discovered an endocrine axis that controls male reproductive activity, connecting the XO-SG, the AG, and the testes in crustaceans [31]. Therefore, the “phototransduction-fly” pathway, which is related to the photoperiod, and is implicated in regulating biological processes, was identified through transcriptome analysis comparing dsM vs. dsRM and dsM vs. dsFM. This pathway has been identified in several studies on female *M. nipponense*. Qiao et al. (2017) analyzed the eyestalks and cerebral ganglia (CG) of *M. nipponense* during both breeding and non-breeding seasons and found that photoperiodic pathways may play a role in regulating crustacean reproduction [32]. In addition, Fu et al. (2019) conducted research on the CG during different stages of ovarian development and proposed that light signals could impact biological clock gene expression in the nervous system and may reduce steroid hormone secretion [33]. Notably, we previously used E_2_ to reverse sex in male *M. nipponense* and identified an important sex reversal pathway, “retinol metabolism”, which is also associated with the eyestalks [22]. In chordates, retinol metabolism plays an important role in the control of cell differentiation, regulation of immune competence, and reproduction during embryogenesis and in the adult organism [34]. In contrast to the studies by Qiao et al. (2017) and Fu et al. (2019), the present study examined samples taken from gonads rather than eyestalks. This suggests that eyestalks may convert a light signal into nerve or humoral signal through retinol proteins, thereby impacting gonadal development and sex differentiation. However, this hypothesis requires further research to clarify the relationship between photoreceptors and sex differentiation through eyestalk removal in juvenile prawns. It is worth noting that the dsM vs. dsRM group was also screened for an opsin gene, long wavelength-sensitive (*LWS*), which is involved in ovarian development and growth [35], and is also thought to play a role in mate-selection-related behavior [36].

In dsFM vs. dsNRM, several pathways are enriched, including lysosomes, steroid biosynthesis, thiamine metabolism, and the longevity-regulating pathway of multiple species. Lysosomes are degradation centers and signaling hubs in cells, and they play important roles in cellular homeostasis, development, and aging [37]. Steroid hormones play essential roles in regulating water and salt balance, metabolism, and stress responses, and in initiating and maintaining sexual differentiation and reproduction [38]. A recent study found that *Cyp17a1* played a crucial role in the production of E_2_ and the determination and differentiation of sex in *Oreochromis niloticus* by affecting the steroid biosynthesis pathway [39]. Thiamine is essential for many physiological functions and is, among other roles, involved in glucose metabolism, the maintenance of the nerve membrane function, and the synthesis of myelin and several types of neurotransmitters (e.g., acetylcholine, serotonin, and amino acids) [40]. The longevity-regulating pathways encompass several genes and associated signaling pathways that can modulate processes such as autophagy, protein synthesis, nutrient sensing, mitochondrial function, and oxidative stress, among others [41]. The final mean weight, WGR, and SGR of dsNRM *M. nipponense* exhibited significantly greater values when compared to normally developing males. Based on these functional pathways and growth data, we posit that *dsIAG* has a significant role in reproduction, as well as the regulation of growth and metabolism, even though it may not completely reverse the sex of certain male *M. nipponense*. Moreover, five growth-related genes, fatty acid synthase (*Fas*), acetyl-CoA carboxylase (*ACC*), delta-9 desaturase (Δ*9D*), glutathione S-transferase (*GST*), and *LWS*, showed different trends in the three comparison groups. These genes are important in the synthesis of fatty acids and cholesterol, and in key regulatory molecules in muscle, ovary, brain, and other tissues [42,43,44,45], which further validates our hypothesis. A study on *grass carp* revealed that an excessive amount of 17α-methyltestosterone (MT) hindered the development of ovaries, which led to a reallocation of energy from ovarian development toward general growth and development [46]. The *IAG* gene plays a crucial role in androgen production, so we posit that elevated body weight was probably due to a decrease in androgen levels, allowing prawns to allocate more energy toward muscle growth and tissue development.

In this study, 18 genes related to reproduction, including 13 sex-related and 5 growth-related genes, were screened in the following groups: dsM vs. dsFM, dsM vs. dsRM, and dsFM vs. dsNRM. In the dsFM vs. dsNRM group, sex-related genes such as vitellogenin receptor (*VGR*), gonadotropin-releasing hormone receptor (*GnRHR*), sperm gelatinase (*SG*), etc. were found to be regulated differently from the other two groups. *VGR* facilitates the secretion of vitellogenin (*VG*) into the hemolymph through acceptor-mediated endocytosis and leads to the transformation of oocytes into yolk proteins during yolkogenesis [47,48]. *GnRHR* is a sex hormone controlling reproductive processes that regulates reproductive functions in vertebrates by stimulating the production and release of pituitary gonadotropins [49,50]. *SG* is a key player in regulating various reproductive functions, including sperm motility (hyperactivation) [51], acrosome reaction [52], and sperm-egg fusion [53]. In male *M. nipponense*, *MnSG* is primarily expressed in the testis and shows a progressive increase in expression as the development of the testis progresses. The expression of cystine (*Cys*) shows inconsistency among the dsFM vs. dsNRM and dsM vs. dsFM groups. *Cys* contributes to male gonad development and also plays a crucial role in oogenesis by inhibiting proteases [54]. *Cys* could play a role in the control of sex reversal by *IAG*, indicating that the molecular regulatory mechanisms underlying sex differentiation are intricate and capable of affecting multiple pathways and genes. This observation supports our previous speculation that in dsNRM, sperm maturation is suppressed, while the ovaries show normal development. However, this discovery prompts an analysis of why these male prawns do not undergo a sex change and become female; or, why the sex reversal does not reach 100%. Previous studies have indicated that sex reversal efficiency in prawns could be influenced by factors such as water temperature, climate, and timing of intervention (injection or feeding), among others [24,55,56]. However, as gonads of the prawns mature, achieving sex reversal becomes increasingly challenging. Nonetheless, we cannot eliminate the possibility of gradually increasing the concentrations of *dsIAG* as a potential solution. We propose that complete sex reversal may depend upon attaining a certain threshold of androgen concentration or the development of specific organs (such as the eyestalks, CG, or gonad) to a particular stage. Under these circumstances, the use of specific stimuli (*dsIAG*, E_2_, MT, etc.) could then facilitate directed gonadal differentiation.

## 4. Materials and Methods

### 4.1. Experimental Design

Thirty healthy holding eggs of female *M. nipponense* (body weight = 4.02 g ± 0.55 g) were obtained from Taihu Lake (Wuxi, China; 120°13′44″ E, 31°28′22″ N) and cultured until they hatched young prawns. As shown in Figure 9, larvae were fed with Artemia until their body weight reached PL30 (0.1278 ± 0.0272 g), and they were assigned randomly into control and experimental groups. The injection was performed every 6 days, and gonad samples were collected on day 30 for tissue observation to determine the interference efficiency and to count the percentage of males. Thereafter, male prawns (dsM), female prawns (dsFM), neo-female prawns (dsRM), and unsex-reversed male prawns (dsNRM) were selected for transcriptome sequencing.

### 4.2. dsIAG Synthesis and Injection

Snap Dragon (https://www.flyrnai.org/cgi-bin/RNAi_find_primers.pl (10 August 2022) was used to design a specific RNAi primer of *IAG* with a T7 promoter site, as described in our previous study (Appendix A). The Transcript Aid T7 High Yield Transcription kit (Fermentas, Inc, Chicago, IL, USA) was used to synthesize the *MnIAG* dsRNA (*dsIAG*), and the purity and integrity of the dsRNA were detected by 1.2–1.5% agarose gel electrophoresis. The dsRNA concentration was determined using a BioPhotometer UV spectrophotometer, adjusted to the desired concentration with DEPC water, and stored at −80 °C.

*M. nipponense* received *dsIAG* (8 μg/g) via injection into the pericardial cavity, whereas the control group was injected with equivalent volumes of DEPC water. To maintain the interference efficacy of dsRNA and minimize prawn mortality, subsequent injections were administered every 6 days following the initial injection for a total of 6 times.

### 4.3. Histological Observations

The morphological changes of the gonads after RNAi treatment were observed by hematoxylin and eosin (HE) staining. Prawn samples of PL10 and PL30 after six injections by *dsIAG* were mounted on slides and compared with normal prawn of the same developmental period [57]. The operation was conducted according to the description of the previous study. Observations were performed using a stereo microscope (SZX16; Olympus Corporation, Tokyo, Japan). The various cell types were labeled following the description of the previous report.

### 4.4. Growth Traits

The mean body weight of juvenile prawn was measured at the beginning of the experiment (initial mean body weight) and *dsIAG* treatment after 30 days (final mean body weight).

The weight growth rate (*WGR*) was determined as follows:(1)WGR=(Wt−W0)/W0×100%

The specific growth rate (*SGR*) was determined as follows:(2)SGR=(In Wt−In W0)/t×100%
where *Wt* is the average body weight 30 days after treatment with *dsIAG*, *W*0 is the average body weight, and *t* is the total number of days of the experiment.

### 4.5. Sex Ratio and Feminization Rate Statistics

The prawn in the control and experimental groups were randomly selected, with more than 90 individuals providing statistics for the sex ratio and feminization rate. Each group had at least three replicates.

### 4.6. RNA Isolation and Quantitative Real-Time PCR Analysis

Total RNA was extracted by homogenizing the gonads with TRIzol reagent (Autolab Tech, Beijing, China). The RNA concentration was measured using a Qubit RNA Kit in conjunction with a Qubit 2.0 Fluorometer (Life Technologies, Carlsbad, CA, USA). Additionally, the purity of the RNA was evaluated utilizing a Nanodrop 2000 spectrophotometer (Thermo Scientific, Waltham, MA, USA). The integrity of the RNA was assessed employing an RNA Nano 6000 detection kit (2100 Bioanalyzer System; Agilent Technologies, Santa Clara, CA, USA).

The gonads RNA was extracted (100 mg) using a 1 mL TRIzol reagent (TaKaRa, Ohtsu, Japan) and first-strand cDNA synthesis was carried out using a Reverse Transcriptase M-MLV Kit (TaKaRa). The qRT-PCR was performed using a Bio-Rad iCycler iQ5 real-time PCR system (Hercules, CA, USA), with eukaryotic translation initiation factor 5 A as the reference gene. Primers used are shown in Appendix A. The reaction was amplified with 35 cycles at 94 °C for 30 s, 50 °C for 30 s, and 72 °C for 1 min, followed by a 10 min incubation at 72 °C as a final extension step. Each sample had four replicates while each reaction had three controls: nuclease-free water, primer-free water, and template-free water. The system recorded fluorescence curve and data automatically, and dissociation curves of the amplified products were analyzed at the end of each PCR. The mRNA expression levels were determined using the 2^−ΔΔCT^ method.

### 4.7. Transcriptomic Sequencing

The gonads (male: testis; female: ovary) were collected from three individuals in each group and immediately stored in liquid nitrogen at a temperature of −190 °C. A sequencing library was prepared using the NEBNext Ultra RNA Library Prep Kit (Illumina, San Diego, CA, USA) following the manufacturer’s instructions, with 3 µg of RNA from each sample. The RNA was purified and fragmented into small random pieces using poly-T oligo attached magnetic beads (Life Technologies). Double-stranded DNA was then synthesized using a TruSeq™ Stranded mRNA Prep Kit (Illumina). Next, the DNA fragments in the library with a length range of 150–200 bp were screened and purified utilizing an Ampure XP system (Beckman Coulter, Beverly, MA, USA). The purified double-stranded cDNA underwent size selection and connection before incubation with 3 mL of USER enzymes (NEB, Ipswich, MA, USA) at 37 °C for 15 min followed by culturing at 95 °C for 5 min. Polymerase chain reaction (PCR) was performed using Phusion High Fidelity DNA polymerase, universal PCR primers, and index (X) Primer. The products were subsequently purified using the Ampure XP system. Finally, the composite samples were paired and sequenced using a HiSeq™ 25,000 for 2 × 100 bp according to the manufacturer’s instructions. Each lane’s PE reading was approximately 150 m (n = 3).

The Illumina HiSeq high-throughput sequencing platform, which utilizes sequencing by synthesis technology, can generate a substantial amount of high-quality raw data. FastQC tools were used to truncate adapter and primer sequences and remove reads with N > 10% and quality (Q) < 5 for > 50% of reads. Reads were assembled using Trinity according to the parametric transcription group, employing a minimum contig length of 300 and K-mer set at 27. Six pairwise comparative sets of DEGs were obtained for dsM, dsFM, dsRM, and dsNRM groups using DESeq2 analysis. The false discovery rate (FDR) was calculated using the Benjamini Hochberg correction method to correct the significance of the *p*-values, with screening criteria of |log2(fold change)| ≥ 1 and FDR < 0.05 used to define DEGs. The pathway enrichment analysis of DEGs was performed using GO, COG, and KEGG annotation methods, with significantly enriched pathways identified based on FDR < 0.05.

### 4.8. Quantification and Statistical Analysis

The statistical analyses were all conducted using IBM SPSS Statistics for Windows, version 23.0. (IBM Corporation, Armonk, NY, USA). The significant differences between groups were determined by one-way ANOVA, followed by the least significant difference and Tukey’s test. Quantitative data were expressed as mean ± standard deviation (SD). Probability (*p*) values < 0.05 were considered statistically significant.

## 5. Conclusions

This study successfully reversed the sex of *M. nipponense* by using *dsIAG* and established gonadal transcriptome libraries for neo-females (dsRM) and unsex-reversed males (dsNRM). The *dsIAG* treatment resulted in neo-female ovaries developing slower than normal female *M. nipponense*. Histological and transcriptional differences in the gonads of dsRM and dsNRM were compared, revealing key metabolic pathways and genes involved in sexual development. It was predicted that the eyestalks and the “phototransduction-fly” photoperiodic pathways of *M. nipponense* play important roles in sex reversal. Furthermore, the pivotal role of *IAG* in reproduction, growth regulation, and metabolism was postulated by integrating DEGs, growth traits, and associated pathways in dsNRM. Finally, we posited that a complete sex reversal may depend on specific stimuli at critical periods. These findings provide an important basis for *IAG* regulation of sex differentiation, reproduction, and growth of *M. nipponense*, as well as establishing conditions for a monoculture.

## Figures and Tables

**Figure 1 ijms-24-14306-f001:**
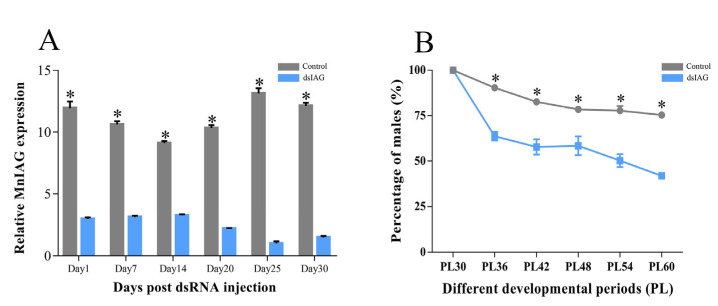
(**A**) Expression characterization of *MnIAG* on different days after *dsIAG* injection; (**B**) The percentage of PL30 male prawn in control and RNAi groups on different days after injection. PL: post-larvae developmental stage. * Indicates significant expression difference between the control and RNAi groups on the same day.

**Figure 2 ijms-24-14306-f002:**
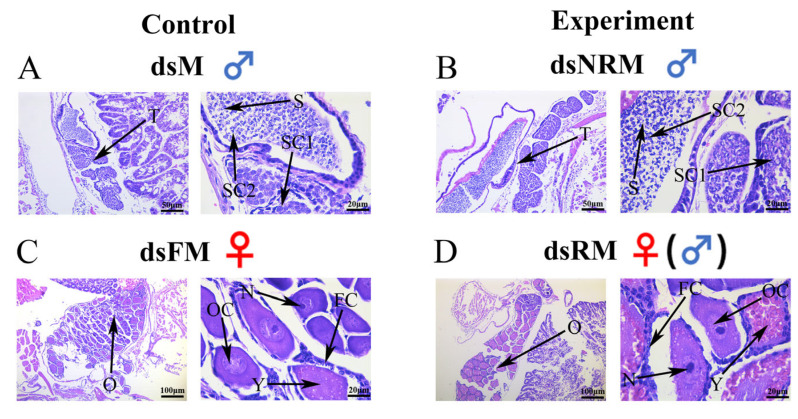
Histological sections of control and RNAi groups at PL30. (**A**) Gonad of normal male; (**B**) Gonad of unreversed male; (**C**) Gonad of normal female; (**D**) Gonad of neo-female. Data are shown as mean ± SD of gonads from separate individuals (n = 3). T: Testis; S: Sperm; SC1: Primary spermatocyte; SC2: Second spermatocyte; O: Ovary; OC: Ovarian cavity; FC: Follicle cells; N: Nucleus; Y: Yolk granule. Scale bars: 20, 50, and 100 µm.

**Figure 3 ijms-24-14306-f003:**
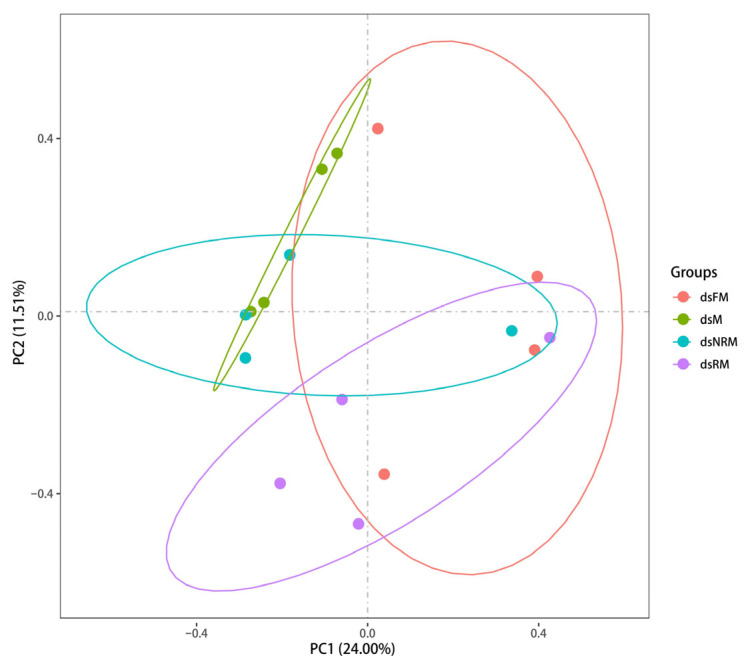
PC1 and PC2 represent the first and second principal components, and the percentages in parentheses represent the contribution of the first principal component to the sample variance.

**Figure 4 ijms-24-14306-f004:**
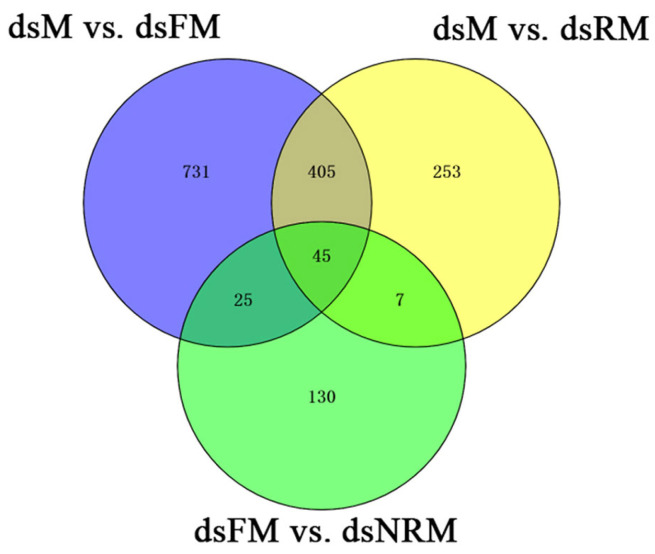
Analysis of DEGs by Venn diagram showing the number of DEGs in M vs. FM, M vs. RM, and FM vs. NRM comparisons.

**Figure 5 ijms-24-14306-f005:**
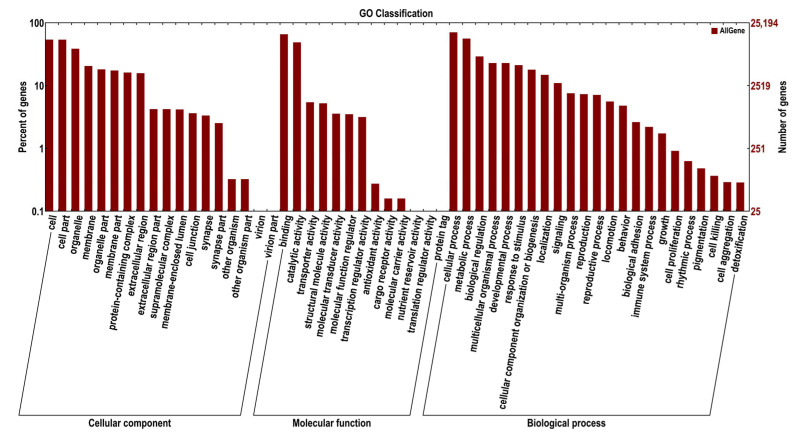
GO classification of the unigenes and DEGs discovered. The abscissa shows the second-level terms in the three GO categories. The ordinates show the number of genes annotated to each term and the percentage of all genes.

**Figure 6 ijms-24-14306-f006:**
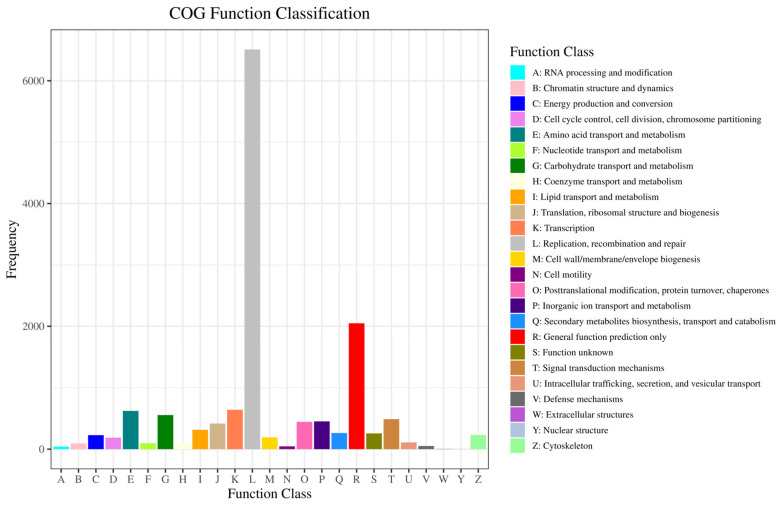
Clusters of orthologous groups of proteins (COG) classification of putative proteins.

**Figure 7 ijms-24-14306-f007:**
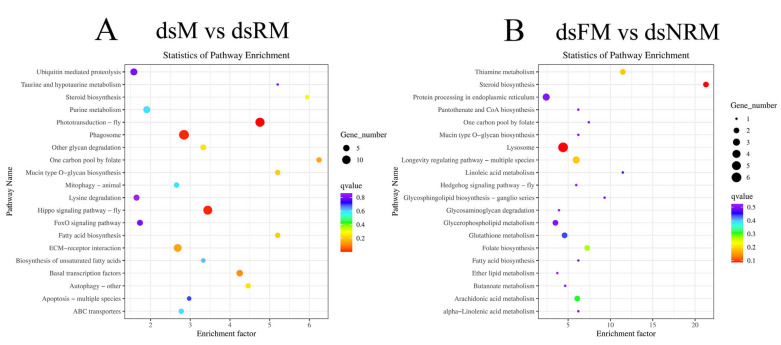
KEGG enrichment of DEGs: (**A**) “dsM vs. dsRM”; (**B**) “dsFM vs. dsNRM”. The size of the dots represents the number of genes. The color of each dot represents the *p*-value. The abscissa shows the enrichment score. The ordinate shows the number of genes annotated to each term and the percentage of all genes.

**Figure 8 ijms-24-14306-f008:**
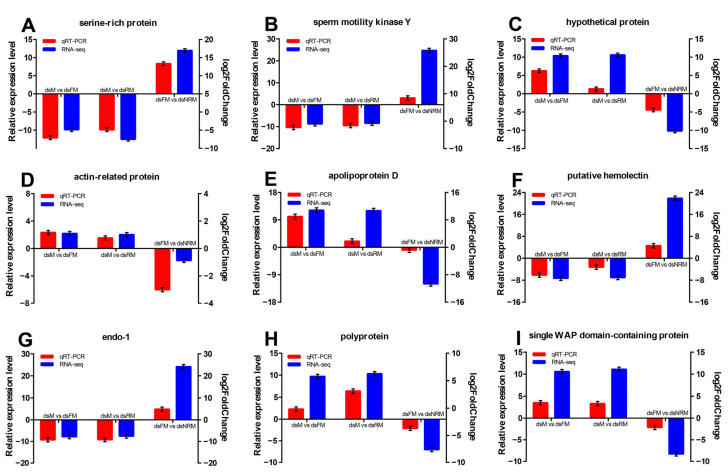
qRT-PCR validation of RNA-Seq data. (**A**) serine-rich protein; (**B**) sperm motility kinase Y; (**C**) hypothetical protein; (**D**) actin-related protein; (**E**) apolipoprotein D; (**F**) putative hemolectin; (**G**) endo-1; (**H**) polyprotein; (**I**) single WAP domain-containing protein.

**Figure 9 ijms-24-14306-f009:**
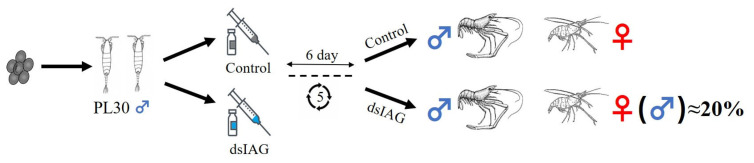
Experimental design.

**Table 1 ijms-24-14306-t001:** Effects of *dsIAG* on growth performance of juvenile prawn at PL30.

Group (μg/g)	Sex	Index			
		IMW (mg)	FMW (mg)	WGR (%/g)	SGR (%/d)
Control	Male	129.85 ± 1.90 ^a^	262.57 ± 9.49 ^a^	50.52 ± 1.82 ^a^	2.34 ± 0.12 ^a^
	Female	141.41 ± 1.98 ^A^	353.37 ± 19.52 ^B^	60.02 ± 2.16 ^A^	3.06 ± 0.18 ^A^
DsIAG	Neo-female	112.32 ± 2.64 ^A^	265.40 ± 17.28 ^A^	57.56 ± 2.86 ^A^	2.86 ± 0.22 ^A^
	Unsex-reversed male	112.32 ± 2.64 ^a^	305.93 ± 6.82 ^b^	63.28 ± 0.83 ^b^	3.34 ± 0.07 ^b^

Note: The data are expressed as mean ± SD (n = 30). IMW: initial mean weight. FMW: final mean weight. WGR: weight growth rate. SGR: specific growth rate. Lowercase letters indicate the significant differences between male and unsex-reversed male, while uppercase letters indicate the significant differences between female and neo-female.

**Table 2 ijms-24-14306-t002:** Quality control and data statistics for clean reads.

Sample	ReadSum	BaseSum	GC(%)	Q20(%)	Q30(%)
dsM1	22,290,216	6,687,064,800	44.04	96.48	90.96
dsM2	21,213,797	6,364,139,100	44.67	96.00	90.02
dsM3	27,395,675	8,218,702,500	43.89	96.51	90.99
dsM4	19,865,327	5,959,598,100	44.42	96.62	91.16
dsFM1	21,757,005	6,527,101,500	43.57	96.34	90.61
dsFM2	24,507,624	7,352,287,200	43.38	96.09	90.19
dsFM3	24,153,175	7,245,952,500	42.94	95.97	89.94
dsFM4	19,776,829	5,933,048,700	43.49	96.19	90.31
dsRM1	18,570,225	5,571,067,500	43.99	96.47	90.90
dsRM2	21,045,097	6,313,529,100	43.33	96.10	90.21
dsRM3	21,327,753	6,398,325,900	44.41	96.43	90.78
dsRM4	25,840,035	7,752,010,500	43.60	96.04	90.07
dsNRM1	21,015,809	6,304,742,700	44.41	96.15	90.24
dsNRM2	21,060,098	6,318,029,400	45.14	96.50	91.01
dsNRM3	21,433,329	6,429,998,700	43.24	96.54	91.05
dsNRM4	20,540,615	6,162,184,500	44.69	96.35	90.69

**Table 3 ijms-24-14306-t003:** Key differential expression pathways in “dsM vs. dsRM”.

No.	Pathway	Pathway ID	dsM vs. dsFM		dsM vs. dsRM	
			*q*-Value	DEGs Number	*q*-Value	DEGs Number
1	Phototransduction-fly	map04745	0.989	4	0.000	12
2	Hippo signaling pathway-fly	map04391	1.000	4	0.012	11
3	Phagosome	map04145	0.905	10	0.012	14
4	ECM-receptor interaction	map04512	0.905	6	0.145	8

**Table 4 ijms-24-14306-t004:** Key differential expression pathways in “dsFM vs. dsNRM”.

No.	Pathway	Pathway ID	dsM vs. dsFM		dsFM vs. dsNRM	
			*q*-Value	DEGs Number	*q*-Value	DEGs Number
1	Lysosome	map04142	0.989	8	0.086	6
2	Steroid biosynthesis	map00100	0.940	1	0.086	2
3	Thiamine metabolism	map00730	0.535	3	0.188	2
4	Longevity regulating pathway-multiple species	map04213	0.905	4	0.188	3

**Table 5 ijms-24-14306-t005:** Expression of reproduction-related genes in the transcriptome.

No.	Name	Accession Number	Up or Down		
			dsM/dsFM	dsM/dsRM	dsFM/dsNRM
Sex-related genes					
1	vitellogenin	AJP60219.1	up	up	down
2	vitellogenin receptor	AJP60220.1	up	up	
3	VASA-like	AEQ19569.1	up	up	
4	cyclin B	ADB44902.1	up		
5	Fem1b	ANN47504.1	up		
6	ferritin	QDA69873.1	up	up	
7	gonadotropin-releasing hormone receptor	AHB33640.1	up	up	
8	cystatin	AXS76129.1	up	down	
9	doublesex and mab-3 related transcription factor	QDE10512.1	down	down	up
10	heat shock protein	QCC72758.1	down	down	up
11	sperm gelatinase	AFM38794.1	down	down	
12	Kazal-type protease inhibitor	AEW24505.1		down	
13	male reproductive-related protein	ABQ41234.1		down	
Growth-related genes					
1	fatty acid synthase	QDK64693.1	up	up	
2	acetyl-CoA carboxylas	ALK82309.1	up		
3	delta-9 desaturase	AMQ48727.1		up	
4	long wavelength sensitive opsin	ASS36969.1		up	
5	glutathione S-transferase	AGJ70295.1			up

## Data Availability

The data presented in this study are available on request from the corresponding author for scientific purposes.

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
