# Peer review of "Insulin-like Androgenic Gland Hormone Induced Sex Reversal and Molecular Pathways in Macrobrachium nipponense: Insights into Reproduction, Growth, and Sex Differentiation"

_ijms, 2023, doi:10.3390/ijms241814306_

Round 1

Author Response

Dear Reviewers:

We thank the reviewers for their time and effort spent to critically review our manuscript. Based on these comments and suggestions, we have made careful modifications on the manuscript. Below, we attached a point by point response to all questions and concerns.

Responses to the Comments from Reviewer 1

Thanks for reviewer’s comment. This manuscript has been edited by a native English speaker. The manuscript has been thoroughly reviewed for spacing, punctuation, grammatical and spelling errors. We hope that the revision will be able to reach the publication level of the journal.

Minor editorial comments

  1. Line 65: eyestalks were should be eyestalks is

Response: Thanks for reviewer’s comment. It has been revised in line 64.

  1. Line 93: Gonads should be Gonad

Response: Thanks for reviewer’s comment. This section has been deleted.

  1. Line 138: add space after day

Response: Thanks for reviewer’s comment. It has been revised in line 89.

  1. Line 140: delete % after 36, 7.68

Response: Thanks for reviewer’s comment. It has been revised in line 91.

  1. Lines 154, 155: normally should be normal

Response: Thanks for reviewer’s comment. It has been revised in line 126.

  1. Line 239: add (GnRHR) after receptor

Response: Thanks for reviewer’s comment. It has been revised in line 215.

  1. Line 261: Significantly should be deleted or revise the sentence

Response: Thanks for reviewer’s comment. This section has been deleted.

  1. Line 274: 17-Estradiol should be 17-estradiol

Response: Thanks for reviewer’s comment. It has been revised in line 250.

  1. Line 280: differentially expressed genes (DEGs) should be DEGs

Response: Thanks for reviewer’s comment. It has been revised in line 255.

  1. Line 294: Khalaia should be Karplus et (2003)

Response: Thanks for reviewer’s comment. It has been revised in line 268.

  1. Line 300: add et (2017) after Qiao

Response: Thanks for reviewer’s comment. It has been revised in line 274.

  1. Line 303: add et (2019) after Fu

Response: Thanks for reviewer’s comment. It has been revised in line 277.

  1. Line 310: Qiao et (2017) and Fu et al. (2019)

Response: Thanks for reviewer’s comment. It has been revised in line 284.

  1. Line 322: an essential should be essential (delete an”

Response: Thanks for reviewer’s comment. It has been revised in line 296.

  1. Line 323: role should be roles

Response: Thanks for reviewer’s comment. It has been revised in line 297.

  1. Line 343: 17-Methyltestosterone should be 17-methyltestosterone

Response: Thanks for reviewer’s comment. It has been revised in line 317.

  1. Line 352: (GnRH, should be (GnRHR

Response: Thanks for reviewer’s comment. It has been revised in line 325.

  1. Line 355: yolkogenesis ?

Response: Thanks for reviewer’s comment. We reviewed the relevant literature, and it seems that no problem with this formulation.

  1. Line 383: culture should be cultured

Response: Thanks for reviewer’s comment. It has been revised in line 356.

  1. Line 391: add city, state after Inc,

Response: Thanks for reviewer’s comment. It has been revised in line 370.

  1. Line 392: delete % after 2

Response: Thanks for reviewer’s comment. It has been revised in line 371.

  1. Line 462: add Ohtsu before Japan

Response: Thanks for reviewer’s comment. It has been revised in line 405.

  1. Line 443: delete Calsbad, CA, USA

Response: Thanks for reviewer’s comment. It has been revised in line 422.

  1. Line 460: differentially expressed genes (DEGs) should be DEGs

Response: Thanks for reviewer’s comment. It has been revised in line 438.

  1. References: journal names should be full? See lines 519-520, 559, 588, 659- 660
  2. Titles should not be large capiitals: see lines 521-522, 527-529, 555-556, 575- 576, 582-584, 598, 601, 607-608, 624, 632-633
  3. Lines 522, 540: PLoS

  1. Line 630: delete The before Journal
  2. Lines 636, 640, 682: journal names: the first letter of each word should be large capital
  3. Scientific name of sunfish should be italic in line 679

Response: Thanks for reviewer’s comment. Problems in the reference section have all been revised in line 487-664.

Reviewer 2 Report

Comments about the manuscript:

“Insulin-like Androgenic Gland Hormone Induced Sex Reversal and Molecular Pathways in Macrobrachium nipponense: Insights into Reproduction, Growth, and Sex Differentiation”

The androgen gland of male crustaceans plays a crucial role in regulating sexual differentiation and maintaining secondary sex characteristics. Insulin-like androgenic substances would be responsible for the stimulation of these glands. A previous study successfully induced sex reversal in Macrobrachium rosenbergii using dsRNA IAG. The study presented here examines the effects of dsIAG (double-stranded insulin-like androgen glandular hormone) on sex reversal in Macrobrachium nipponense, investigating the molecular mechanisms involved in sex differentiation and reproduction in crustaceans.

This well-conducted and well-written work should be able to be published after, however, some improvements to the manuscript.

Page 2, line 53: “The oriental river prawn (Macrobrachium nipponense) [19,20] is a sexually dimorphic crustacean widely distributed in China, Japan, and Korea, with a value of over 20 billion annually”: 20 billion enough to? Animals? Dollars? thank you to specify

Page 2, line 61: “The gonads of M. nipponense begin to develop at PL10 (PL: post-larvae developmental stage)”: according to which table was the stage of development determined? A reference would be welcome.

Page 2: at the end of the introduction, it would be useful to indicate what is the interest of sexual inversion in this species.

Page 13, line 382. “Healthy pregnant female M. nipponense”: how many individuals were used in total?

Page 13, line 399, Histological observations: give details about the histological study: what were the fixative, the embedding medium (paraffin, waxes?), the method of dehydration before embedding, the slice thickness, the supplier of the products with references, the mounting medium...

Page 14, line 403:” Operation according to the description of the previous study”: even if the operation has been used before, give a brief description here.

Page 14, lines 420-421. “The RNA concentration was measured using a Qubit RNA Kit in conjunction with a Qubit 2.0 Fluorometer”: give a brief description of the method used here.

Page 14, line 436 “using the 2−ΔΔCT method.”: briefly explain the method.

Author Response

Dear Reviewers:

We thank the reviewers for their time and effort spent to critically review our manuscript. Based on these comments and suggestions, we have made careful modifications on the manuscript. Below, we attached a point by point response to all questions and concerns.

Responses to the Comments from Reviewer 2

Major comments

Page 2, line 53: “The oriental river prawn (Macrobrachium nipponense) [19,20] is a sexually dimorphic crustacean widely distributed in China, Japan, and Korea, with a value of over 20 billion annually”: 20 billion enough to? Animals? Dollars? thank you to specify

Response: Thanks for reviewer’s comment. It has been revised in line 53.

Page 2, line 61: “The gonads of M. nipponense begin to develop at PL10 (PL: post-larvae developmental stage)”: according to which table was the stage of development determined? A reference would be welcome.

Response: Thanks for reviewer’s comment. It has been added in Table S1 in line 62.

Page 2: at the end of the introduction, it would be useful to indicate what is the interest of sexual inversion in this species.

Response: Thanks for reviewer’s comment. It has been added in line 54.

Page 13, line 382. “Healthy pregnant female M. nipponense”: how many individuals were used in total?

Response: Thanks for reviewer’s comment. It has been revised in line 355.

Page 13, line 399, Histological observations: give details about the histological study: what were the fixative, the embedding medium (paraffin, waxes?), the method of dehydration before embedding, the slice thickness, the supplier of the products with references, the mounting medium...

Page 14, line 403:” Operation according to the description of the previous study”: even if the operation has been used before, give a brief description here.

Page 14, lines 420-421. “The RNA concentration was measured using a Qubit RNA Kit in conjunction with a Qubit 2.0 Fluorometer”: give a brief description of the method used here.

Page 14, line 436 “using the 2−ΔΔCT method.”: briefly explain the method.

Response: Thanks for reviewer’s comment. All methodology-related recommendations have been revised.

Author Response

Dear Reviewers:

We thank the reviewers for their time and effort spent to critically review our manuscript. Based on these comments and suggestions, we have made careful modifications on the manuscript. Below, we attached a point by point response to all questions and concerns.

Responses to the Comments from Reviewer 3

Major comments

  1. The main weakness of the study is the fact that the authors do not have a clear way to check whether their experimental subjects are genetic males or genetic females. The transcriptomic study is very weak since the authors do not know whether the animals used for the different groups of the transcriptomic libraries where actually genetic males or females, thus their differentially expressed genes are not significantly demonstrating an important sexual/biological phenomenon.

Response: Thanks for reviewer’s comment. Sex molecular markers are genetic markers that can be used to determine the sex of an individual. These markers are typically found on the sex chromosomes, X and Y, and can be used in various fields. A sequence of genes for sex identification has been recognized in Macrobrachium rosenbergii. However, unfortunately, this sequence is not applicable to our study species, Macrobrachium nipponense. Therefore, when designing the experimental protocol, we selected male prawn (PL30) that had just developed secondary sexual characteristics. Refer to this article for specific differentiation methods: Sex Reversal Induced by Dietary Supplementation with 17α-Methyltestosterone during the Critical Period of Sex Differentiation in Oriental River Prawn (Macrobrachium nipponense). In addition, we plot the principal component analysis (PCA) when performing transcriptome analysis (Figure 3). PCA was used to calculate the correlation coefficient between different samples and differentiate them from one another. Based on the results of this figure, it is clear that dsM, dsRM and dsNRM are well distinguished. This is another evidence for the success of sex reversal. Finally, neo-females are smaller than normal females, which is abnormal. We found slow gonadal development in neo-females when selecting transcriptome samples. In fact, whole-genome resequencing of neo-females can also help us screen for sex molecular markers in Macrobrachium nipponense.

  1. The article includes two parts that are not necessarily related to each other and are affected differently from the above problematic identification of the experimental groups.

Response: Thanks for reviewer’s comment. The original purpose of adding the study on PL10 Macrobrachium nipponense was to make the overall experimental design more fluid and logical. However, the difference in purpose of these two experimental studies makes the whole thesis of the article a bit strange indeed. Based on your proposal, we removed the section on PL10 and retained only the histological sections, growth traits and transcriptomic studies on PL30 male prawn.

  1. linguistic revision

Response: Thanks for reviewer’s comment. This manuscript has been edited by a native English speaker. The manuscript has been thoroughly reviewed for spacing, punctuation, grammatical and spelling errors. We hope that the revision will be able to reach the publication level of the journal.

  1. Sex reversed males created by IAG-switch manipulation (such as RNAi), are feminized, so become ‘neo females’ and not ‘neo males’

Response: Thanks for reviewer’s comment. All descriptions of sex-reversed individuals in the text have been changed from ‘neo-male’ to ‘neo-female’.

  1. Prawn females are holding their eggs abdominally thus are not ‘pregnant’. Better terms are: ‘holding eggs’, ‘berried’, etc

Response: Thanks for reviewer’s comment. ‘pregnant’ has been changed to ‘holding eggs’.

  1. It is not clear how issues of steroids such as 17β-Estradiol are related to the IAG-switch endocrine system. Also, the term androgens is more related to mammalian steroid systems that were not proven to be the same in prawns.

Response: Thanks for reviewer’s comment. In our previous study, we were also successful in causing sex reversal using steroid hormones (MT, E2), but MT was unable to reverse female prawn that had already developed secondary sex characteristics, which means that breeding neo-females is a viable option. Using hormones in production is strictly prohibited, so we hoped to achieve a similar effect to E2 by inhibiting IAG expression. Although there is currently no evidence that 17β-Estradiol are related to the IAG-switch endocrine system, we did obtain neo-female by two different methods. Therefore, in the Discussion section, we compared these two neo-females in order to identify their similarities and the key pathways that cause sex reversal. In the future, we will conduct functional studies of key sex genes and explore their relationship to the regulation of IAG and steroid hormones.
